# Low-Cost Hyperspectral Imaging to Detect Drought Stress in High-Throughput Phenotyping

**DOI:** 10.3390/plants12081730

**Published:** 2023-04-21

**Authors:** Andrea Genangeli, Giovanni Avola, Marco Bindi, Claudio Cantini, Francesco Cellini, Stefania Grillo, Angelo Petrozza, Ezio Riggi, Alessandra Ruggiero, Stephan Summerer, Anna Tedeschi, Beniamino Gioli

**Affiliations:** 1Department of Agriculture, Food, Environment and Forestry (DAGRI), University of Florence, Piazzale delle Cascine 18, 50144 Florence, Italy; andrea.genangeli@unifi.it (A.G.); marco.bindi@unifi.it (M.B.); 2Institute of Bioeconomy (IBE), National Research Council (CNR), Via Caproni 8, 50145 Florence, Italy; giovanni.avola@cnr.it (G.A.); claudio.cantini@ibe.cnr.it (C.C.); ezio.riggi@cnr.it (E.R.); 3Centro Ricerche Metapontum Agrobios-Agenzia Lucana di Sviluppo e di Innovazione in Agricoltura (ALSIA), S.S. Jonica 106, km 448,2, 75010 Metaponto di Bernalda, Italy; francesco.cellini@alsia.it (F.C.); angelo.petrozza@alsia.it (A.P.); stephan.summerer@alsia.it (S.S.); 4D1 National Research Council of Italy, Institute of Biosciences and Bioresources, Via Università 133, 80055 Portici, Italy; stefania.grillo@ibbr.cnr.it (S.G.); alessandra.ruggiero@ibbr.cnr.it (A.R.); anna.tedeschi@cnr.it (A.T.)

**Keywords:** low-cost hyperspectral camera, high-throughput phenotyping, hyperspectral index, red-edge, optical sensor, drought stress, tomato, projected shoot area, hue, senescence index

## Abstract

Recent developments in low-cost imaging hyperspectral cameras have opened up new possibilities for high-throughput phenotyping (HTP), allowing for high-resolution spectral data to be obtained in the visible and near-infrared spectral range. This study presents, for the first time, the integration of a low-cost hyperspectral camera Senop HSC-2 into an HTP platform to evaluate the drought stress resistance and physiological response of four tomato genotypes (770P, 990P, Red Setter and Torremaggiore) during two cycles of well-watered and deficit irrigation. Over 120 gigabytes of hyperspectral data were collected, and an innovative segmentation method able to reduce the hyperspectral dataset by 85.5% was developed and applied. A hyperspectral index (H-index) based on the red-edge slope was selected, and its ability to discriminate stress conditions was compared with three optical indices (OIs) obtained by the HTP platform. The analysis of variance (ANOVA) applied to the OIs and H-index revealed the better capacity of the H-index to describe the dynamic of drought stress trend compared to OIs, especially in the first stress and recovery phases. Selected OIs were instead capable of describing structural changes during plant growth. Finally, the OIs and H-index results have revealed a higher susceptibility to drought stress in 770P and 990P than Red Setter and Torremaggiore genotypes.

## 1. Introduction

The global population is projected to reach 9.8 billion in the year 2050, generating a growing demand in food production [1,2]. The global agricultural sector is facing growing risks associated with more frequent extreme meteorological events [3,4] that are capable of undermining classic species commonly used in food production. The direct consequence of this phenomenon is a loss in terms of the quantity and quality of production and a threat to food security [5,6,7]. Indeed, from 2008 to 2018, the production loss at regional scale caused by anomalous weather events was estimated at around 8% of the yearly total for an overall value of 116.7 billion USD [8,9,10,11,12]. The need for increased food production comes with an increasing need for global agriculture to adapt to global climate changes, with the final objective of producing high-quality food while alleviating environmental impact and greenhouse gas emissions. All these challenges are crucial for the various stakeholders employed in the agricultural and food production chains, as highlighted by global institutions [13,14,15,16,17].

To improve the food quality and to reduce the environmental impact of crop cultivation, producers can consider different strategies, such as optimizing crop water consumption, optimizing the amount of fertilization used during the cultivation cycle [18] and selecting genotypes that are more resistant to drought, salt and thermal stress [19]. More importantly, the management of food security requires the production of accurate data for crop production in each geographic area and their environmental adaptability at all growth stages [20,21]. The selection of new genotypes capable of withstanding stressful situations due, for example, to unpredictable climatic changes is one of the seed industries’ main challenges. Nevertheless, genetic selection alone is unable to predict the interaction between genotypes and the environment [22]. Phenotypes express the result of such interactions, and their assessment is fundamental to reach satisfactory breeding results. Environmental factors can affect plants development and measuring their effect on genotype expression represents a primary method to discriminate those genotypes that are better suited to be adopted in a specific cultivation area [21,23]. To accomplish those issues, applying non-invasive and low-cost technologies to monitor crops’ development and health status has become essential in plant breeding and phenotyping techniques [24].

High-throughput plant phenotyping (HTP) can be considered a rapid, efficient and low-cost tool to test the interaction between genotypes and the environment across a large number of genotypes. The use of high-throughput plant phenotyping platforms (HTPPs) is a powerful way to provide information on the plant growth process [25] through the measurement of biophysical traits, such as biomass, dry and wet weight, leaf area index (LAI) [26,27] and transpiration (Tr) [28]. Classical biometrical measurements can be integrated with optical indices (OIs) based on colour variation to retrieve information about the plant life cycle. Normally, the acquisition process consists of multiple image acquisitions using the red, green and blue colour model (RGB) combined with the collection of biometrical parameters over time [22].

Since the early 2000s, hyperspectral imaging (HI) has been used in scientific research, for example in environmental monitoring, vegetation analysis, atmospheric characterization, and biological and chemical detection. HI is a non-invasive technique based on the proprieties of light reflectance in the VIS/NIR/SWIR spectral range, and the application of HI techniques is widely used by scientists to retrieve information about the biochemical and the biophysical properties of crops [29,30]. In the last 20 years, hyperspectral sensors have been successfully applied at different spatial scales of investigation, e.g., at the satellite scale, to obtain the ecosystem properties over large areas (at 10–30 m grid spatial resolution) during research activities conducted by the European Space Agency (ESA) and Italian Space Agency (ASI) [31,32]; at the landscape scale through aerial surveys, to retrieve field level spectral data at 0.5–5 m grid spatial resolution), such as in a field drought stress experiment conducted on *Zea mays* plants by Damn et al. [33]; at the plant scale (at centimeters spatial resolution) to investigate the vineyard health status using an unmanned aerial vehicle (UAV) [34]; and at the proximal scale, collecting information from the inner structure of fruits (at sub-centimetric spatial resolution), as in the case of the early detection with a non-invasive method to detect the moldy core infection in apple fruits [35].

New approaches in the HTP field also include the use of HI. Therefore, HI’s ability to classify a plants’ stress status using a non-invasive method is attractive for HTP applications [36]. Researchers have used several applications of HI in HTP and HTPP during phenotyping experiments, e.g., to retrieve spectral vegetation indices (VIs), to estimate plant biophysical and biochemical traits [37] through the ratio of VIS/NIR/SWIR spectral bands, or for the early detection of plant diseases, e.g., Thomas et al. [38] successfully introduced a new HI phenotyping system in a greenhouse experiment, which creates a field-like situation for the early detection of powdery mildew. However, the use of HI in full-scale phenotyping applications remains a distant goal due to the cost of the instruments, the significant time required for data processing, the large amount of memory occupied by hyperspectral data and the challenge of developing rapid and efficient image segmentation processes [29,37,38,39,40]. In recent years, a new generation of portable and low-cost HI cameras and sensors with a VIS/NIR spectral range have been developed and made commercially available. Such devices are relatively straightforward in producing an image composed of multiple spectral bands, thus increasing their possible use in creating new VIs for plant phenotyping applications [29]. Studies concerning the application and test of low-cost HI technologies in a HTPP are therefore urgently required to develop new, fast and straightforward segmentation methods, new hyperspectral indices to detect crops status, and to develop and test new phenotyping methodologies alternative to commonly adopted OIs [41,42].

The current work is based on the integration of the Senop HSC-2 low-cost hyperspectral camera (HSC-2) [34] in the HTPP based on Lemna Tec Scanalzer3D system (LemnaTec GmbH, Aachen, Germany), a node of the European Plant Phenotyping Network (EPPN), during a tomato drought stress experiment conducted from May to July 2021. The agronomic and physiologic responses of four tomato genotypes (770P, 990P, Red Setter and Torremaggiore), characterized by a high level of industrial interest, were tested under two cycles of well-watered and deficit irrigation. Several optical indices (OIs) derived by HTPP imaging analysis, such as projected shoot area (PSA) [43], hue index (HUE) [44], and colour senescence index (SI) [45], obtained during the image acquisition process, were analysed and discussed. Subsequently, the OIs were compared with the HI results derived by the hyperspectral acquisition with the HSC-2. Therefore, the work aims to evaluate the improvement achieved using a low-cost portable hyperspectral camera in an experiment carried out in a high-throughput plant phenotyping platform (HTPP) when monitoring the water stress tolerance of tomato genotypes.

## 2. Results

### 2.1. Environmental Variations and ET

Relative humidity (RH) and temperature (T) data were collected within the HTP greenhouse environment for the duration of the experiment. The vapour pressure deficit (VPD) was computed from the start to the end of the experiment (Figure 1).

The daily maximum RH (RHmax) was registered on 24 May, and the daily minimum RH (RHmin) was registered on 14 June (Figure 1a). Air temperature showed a general increase over time. The daily maximum temperature (TMax) was registered on day 20 June, and the daily minimum temperature (TMin) on day 22 May (Figure 1b). RH and T show opposite trends during the initial stages of the experiment (until 4 June). Subsequently, the two trends appear similar from 5 June to 11 June, while they returned opposite from 12 June to the end of the experiment. VPD values spanned from 1 kPa to 2.3 kPa and followed a general increment reflecting the temperature rise during the experiment (Figure 1c).

The evapotranspiration (ET) was computed for all four genotypes and for all water treatments by daily weight differences between pots (Figure 2).

The first stress phase (27 May–2 June, Figure 2) exhibits ET differences, where 770P and 990P show higher differences in ET than Torremaggiore and Red Setter between control and stressed theses. ET grows during the first stress phase in all the watered thesis, likely reflecting the plant growth process and the consequent increase in leaf area and evapotranspiration favored by water availability in the soil. During the first recover phase (2–8 June, Figure 2) ET values remain overall stable in the watered thesis, while they increase in the stressed theses, converging to similar values between control and stressed thesis for all genotypes on 8 June 2021, and concomitantly to the end of the first recovery period. This pattern highlights an efficient recovery in all genotypes. The second stress phase (8–15 June 15, Figure 2) exhibits higher levels of ET, and the total amount of water lost reached peaks of 900 g per day in 770P and 990P genotypes, higher than Torremaggiore and Red Setter (800 and 750, respectively).

### 2.2. Hyperspectral Analysis

#### Segmentation Process

Over 250 HI images were collected for a total amount of 120.6 Gigabytes. A segmentation process was adopted to obtain an optimal separation between plants and background (Figure 3) for all hyperspectral images in a straightforward mode that is suitable for high-throughput applications. The segmentation was based on selecting the reflectance pixels above a specific percentile threshold that was set at 96.

Before the segmentation process (a), the image was composed of both the background and the plant pixels, including illuminated and shaded areas, as well as a set of different leaf geometries given the three-dimensional plant structure; after the segmentation process (b), the image was composed of only plant regions with a reflectance value larger than the value of the 96° percentile, with a total amount of 11408 pixels belonging to the plant region in this specific case (Figure 3). The results show a clear separation between plants and background. The spectral band selected to extrapolate the plant region from the background was identified at 750 nm in the NIR spectral region corresponding to the end of the red edge curve (Figure 4). The relatively high reflectance of vegetation in this spectral region allowed for the leaf regions to be correctly isolated, receiving a direct and orthogonal illumination from shaded areas or areas affected by an oblique illumination geometry.

### 2.3. Hypespectral and RGB Cameras Derived Indices

Three optical indices, PSA, HUE and SI, and the H-index were computed from the start to the end of the experiment. The ANOVA analysis was implemented to assess the significance of differences between control and the treated thesis for all the parameters and all the experimental phases (Table 1).

At the end of the first stress period (Table 1), only the hyperspectral index (H-index) and the plant-morphology-related index (PSA) were able to discriminate the differences between the two irrigation treatments within each of the studied genotypes. 

For all the studied genotypes, the stressed plants reported significantly lower H-index values when compared to well-watered plants (Figure 5).

For PSA, significant effects have been ascertained in relation to water stress and the observed differences between water treatment ranging from −45% to −54% (data not shown) in Red Setter and 990P, respectively, emphasizing the negative effect of drought stress, reducing projected shoot areas in stressed plants. PSA values remained significantly lower in all the following intervals for all the studied genotypes except Red Setter (Figure 6).

Both the other colour-related indices (HUE and SI) did not reveal any differences between irrigation treatments after the first week of water stress.

After the end of the first recovery period, as expected, the morphology-related index (PSA) confirmed the ability to discriminate between water treatments for all the genotypes (Table 1), even though, for all the stressed plants, an increment in the index was observed that was higher than that reported for control level. The PSA variation during the first recovery period ranged from +33% to +75% for well-watered and stressed plants, respectively, averaging the increments computed for each genotype (data not showed). 

On the other hand, for light-related indices, the first recovery phase showed a substantial recovery in stressed plants, which exhibited a similar spectral response to the well-irrigated.

In particular, for HI (Figure 5), the increments observed for all the stressed plants in all the studied genotypes led to no significant differences between water treatments, indicating their ability to detect the recovery process.

The values registered for HUE and SI were only shown to be affected by water stress for 990P.

After the second stress period, no significant differences between treatments were ascertained for HI in all the genotypes. However, the projected area showed the ability to discriminate the stressed plants, reporting significantly lower PSA values even though, for Red Setter plants, the measures were not significantly different (Figure 6).

Both colors-related indices (HUE and SI) maintained the same trend of variation (Figure 7 and Figure 8) compared to previous measures (decreasing values for HUE and increasing for SI), but different water stress effects were ascertained. For HUE, significant differences were only measured between water treatments in 990P. However, SI values were lower in stressed plants and this difference was only not significant for 770P.

At the end of the experiment, when the second recovery period was completed, significant differences between water treatments were ascertained only for PSA (except in Red Setter) and in Torremaggiore for the senescence index.

## 3. Discussion

ET results showed a net separation between stressed and control theses according to the differential irrigation regime conducted during the experiment. The second stress phase (8–15 June) registered the maximum levels of ET for all genotypes compared to the other experimental phases, concomitant with the high-temperature peaks registered during the second stress. A consequence of the increased temperature was an increment of VPD during the second stress phase, causing a greater ET demand compared to the first stress phase, in agreement with the results obtained by Noh et al. and Sadok et al. [46,47] who demonstrated the link between VPD and ET. ET differences between stressed and control theses appear more contained compared to trends reported during the first stress. There are few studies in the literature about the drought adaptability of genotypes used in this experiment; depsite this, the results obtained can be explained by a physiological adaptation to the mild drought stress conditions in partially irrigated plants, in accordance with Jureková et al. [48] and Santos et al. [49], who measured the metabolic and physiological changes associated with the adaptation of plant cells to water stress. Overall, the four genotypes show different responses in terms of ET between the two theses due to the stress phases. Genotypes 770P and 990P show greater values of ET in the control theses, showing a lower water stress resistance than Red Setter and Torremaggiore genotypes. ET differences between tomato genotypes can be due to their water stress resistance, expressed in different morphological traits, such as leaf rolling, leaf orientation, leaf size and plant habit [50]. The biomass production and growth rate in tomato plants were influenced by the water availability, and differences in PSA between control and stressed theses can be due to the different irrigation levels, in accordance with the results obtained by Santos et al. and Patanè et al. [49,51], who demonstrated the direct proportionality between available water and biomass production. Red Setter and Torremaggiore show minor differences between stressed and well-irrigated theses in terms of PSA compared to 770P and 990P genotypes. In addition, Red Setter registered lower PSA differences between theses over stress and recovery phases, showing greater tolerance to the water stress condition. Nevertheless, differences in PSA between control and stressed theses are influenced by plant characteristics, such as habit and growth rate, and considering only PSA differences will not describe the genotype’s drought stress susceptibility. HUE results show a change in shade from green to yellow in all genotypes and theses over the experimental phases. The colour change in leaves is a natural physiological effect due to the tomato plant’s senescence and consists of a chlorophyll breakdown from the leaf margin to the interior of the leaf blade, as reported by Quirino et al. [52]. The stress condition causes a decrease in the water content in stressed plant leaves, as measured by Khan et al. [53], and the result of this process is a slowdown in the chlorophyll breakdown. Therefore, a slowdown in a colour change can be attributed to the stress status, in accordance with the results obtained by Janni et al. [54], who conducted a tomato drought stress experiment using the LemnaTec Scanalzer 3D system. The HUE results do not efficiently describe the ET trend, except in the case of the 990P genotype, which reported significant differences between theses at the second stress and second recovery phases.

The most significant variations in SI between control and stressed theses were reported in the 990P genotype, as confirmed by ANOVA results at the end of the first stress and second recovery; at the same time, Torremaggiore showed limited variations in SI in all experimental phases except at the second stress, where the theses are statistically diffracted. SI, as HUE, is a colourimetric index based on the analysis of RGB images. Higher SI values correspond to a light green/yellow plant colour and high SI values in controls are due to their minor ratio between chlorophyll and water compared to stressed theses, where the minor water content increases the chlorophyll and water ratio, showing a green colour that is more intense than that of the control theses [55]. The inconsistencies between HUE, SI and PSA results can be attributed to a differential response over time to the water stress condition between biometric and RGB parameters. Indeed, the slowing of the growth rate is immediately registered in the presence of limited water availability, while colour variation appears with a delay and a less homogeneous distribution between plants.

The segmentation developed technique permitted the minimization of the geometric light-scattering effects in combination with removing the background pixels and shadow effects caused by the complex canopy structure. The light reflection effects due to the overlapped leaves represent a critical issue in obtaining a clean spectral signal, which would require complex 3D radiative transfer models to be resolved. Instead, simplified segmentation approaches such as the one used here are adopted to automatically extract specific leaf subregions [29,36]. Using a homogeneous background in the acquisition process, characterized by low reflectance values compared to the target, represents a fundamental element in obtaining a rapid and efficient segmentation, solving the needs of storage space and speed in the hyperspectral data processing [39,40]. The storage of the HI is an expensive process in terms of memory and using an efficient segmentation method can reduce the occupied memory. After the segmentation, the hyperspectral data were reduced to a total storage space from 120 to 18.7 gigabytes, obtaining a data reduction of 85.5%. Indeed, the application of HI for plant phenotyping in an industrial context needs to be economical in terms of occupied memory and image segmentation efficiency because the application of complex and expensive processes represents an operational bottleneck and finally an increase in cost for the industries.

The trend of the spectral signatures collected during the experiment clearly reflects the presence of vegetation targets with an increase in reflectance from 650 to 720 nm, as reported elsewhere [29,32]. The H-index used here is based on the maximum slope of the spectral reflectance curve in the red edge spectral region. The link between the slope of the red edge spectral region and water stress condition was investigated by several authors, e.g., Boochs et al. and Schlemmer et al. [55,56], who demonstrated the link between the lower slope in the red edge spectral region and the water stress. Using such an index also has a practical advantage associated with the use of low-cost cameras. It is well known that low-cost spectral cameras can have relevant spectral drift due to the sub-optimal sensor quality or thermal stabilization of the sensing elements, resulting in a temperature-dependent spectral shift [34,57,58]. By computing the slope of the reflectance curve, the effect of a potential spectral shift is avoided, only requiring that the spectral shape is maintained. On the other hand, multispectral cameras with a limited number of spectral bands are likely not capable of retrieving an index that requires the red edge spectral shape to be adequately sampled to measure its slope variability. Differences between theses are linked to the different plant’s spectral responses in full and partial irrigation regimes. Indeed, the high H-index value in the control theses indicates a higher slope at the red-edge spectral region than the stressed theses. Compared to the HUE and SI results, the H-index better describes the differential irrigation regimes conducted during the first stress phase. HUE and SI do not show clear differences between theses in the first stress phase, while the H-index clearly describes differences between the two theses in all genotypes. Among all the indexes tested here, H-index was the only index capable of detecting both the first stress phase, leading to significant differences in all theses, and the first recovery leading to non-significant differences in all theses (Table 2), where PSA reported significant differences in all theses and SI and HUE reported significant differences between theses in 990P genotype. This result highlights HI’s greater potential for the early detection of water stress compared to the RGB methods. The optical response in the visible spectral region was less rapid than the near-infrared response. Steidle et al. [59] also found that near-infrared reflectance was capable of providing estimates of leaf water content. The non-significant ANOVA results obtained in the second stress phase could be due to the stressed plants’ water stress adaptability, which showed a similar optical response between theses in the VIS/NIR spectral range [52,53]. The major values of standard deviation in 990P and Red Setter genotypes at the end of the first recovery compared to the first stress can be attributed to the different growth rate between samples. Nevertheless, the global trend of the H-index was not significantly influenced, providing clear results in terms of trend. Overall, 770P and 990P theses show the most significant statistical differences over the experimental phases, particularly when considering the ET results compared to Red Setter and Torremaggiore, as highlighted by PSA, the H-index and, less clearly, by HUE and SI. Therefore, this result indicates the minor capacity of 770P and 990P stressed theses to maintain a similar behaviour to the controls. Consequently, 770P and 990P show a higher susceptibility to water stress than the Red Setter and Torremaggiore genotypes.

## 4. Materials and Methods

### 4.1. Experimental Design

The experiment was conducted in the HTPP located in ALSIA (Agenzia Lucana di Sviluppo ed Innovazione in Agricoltura, S.S. Jonica Km 448,2, 75012 Metaponto MT—40.392217 N 16.788198 E) and based on a LemnaTec Scanalzer 3D system. The HTPP is included in the Italian Plant Phenotyping Network Phen-Italy (Phen-Italy). Phen-Italy is the Italian node of EMPHASIS, the European infrastructure (ESFRI) on plant phenotyping (https://emphasis.plant-phenotyping.eu/, accessed on 15 December 2022). Four tomato genotypes, 770P, 990P, Red Setter and Torremaggiore (Table 2), were tested in a drought stress experiment, from the beginning of May until the end of June 2021. The experiment timetable is reported in Table 3.

Plants were grown in 3.2-litre pots containing 1.8 kg sand–peat mixture, located on the conveyor belt and fully irrigated until 3 weeks (27 May) after transplanting. Then, drought stress was imposed, setting two subsequent cycles, each obtained combining a weekly period of irrigation reduction to 70% of daily ET restoration (stress period) followed by a weekly well-watering period (recovery period). Fully irrigated treatment (100% of daily ET restoration) was adopted as control. Each treatment was replicated six times. The water amount, restored daily, was gravimetrically measured by a balance and then applied by an automatic irrigation control system, both installed on the pot conveyor belt. The daily water weight loss from each pot was used to estimate ET, defined as the water transpired by plants summed to water evaporated daily from pots [60] and computed as follows:ET = TW_d_ − TW_d+1_(1)
where TW_d_ is the weight of the pot measured at the same time each morning, and TW_d+1_ is the pot weight the day after. 

### 4.2. Phenotyping Platform and RGB Acquisition Process

HTPP is based on a LemnaTec Scanalzer 3D system (Figure 9) and consists of an automated belt conveyor system with a tracking system based on bar code and RFID for identifying single plants. Four sequential cameras were installed in the HTPP to take 3D images of plants in the RGB spectral range. The platform allows for the quantitative, non-destructive analysis of different crops or model plants under high-throughput conditions. Each plant was imaged sequentially in multiple scanalyzer3D camera units, and tomato plants were automatically conveyed to the imaging chamber equipped with RGB KAI 2093 image sensor with a 1920 × 1084-pixel (2 megapixels) resolution. Three images were acquired per plant: one from above the plant in the top view (TV) and two from the side view (SV) at 0° and 90°. The plants were illuminated by halogen lamps with a maximum level variation of 2% and a net power of 35 Watts. Spectral acquisitions were carried out simultaneously with RGB acquisitions.

### 4.3. Environmental Monitoring

Environmental conditions were measured every 15 min from the start to the end of the experiment by two RH and T sensors placed on the conveyor belt at the height of 1 m. VPD [46] is the difference between the amount of moisture in the air and the amount the air can hold at saturation, and was obtained using RH and T data as follows:VPD = es − ea(2)
where es is the saturation vapour pressure or vapour pressure measured in kilopascal (kPa) at air temperature, and ea is the actual vapour pressure or vapour pressure measured in kPa at dewpoint temperature. Ea increases or decreases non-linearly. Therefore, the mean of the ea at the mean daily maximum and minimum air temperatures was used for a given period.

### 4.4. Hyperspectral Data Acquisition and Processing

Hyperspectral data were collected using the HSC-2 (Senop Oy, Finland), a low-cost hyperspectral portable snapshot camera equipped with two CMOS sensors [34,58]. The HSC-2 spectral range was set from 650 to 820 nm using an image frame size of 1024 × 1024 pixels (1 megapixel). The spectral resolution was settled around 2.5 nm. A single hyperspectral image was saved as a 3D cube, also called a hypercube (HC), formed by 69 spectral bands overlapping on a single object for a virtual memory of 400 megabytes per image. The exposure time was settled at 80 milliseconds per band, and the distance from the hyperspectral camera to the target was established at 1.5 m (Figure 10). A white Lambertian reference target at 75% of reflectance, capable of reflecting light uniformly [61], was placed in the target area below.

The spectral data were acquired in digital number units and converted in radiance (mW/nm·sr·m²). The HC was converted in reflectance as follows:Ref = Rad/ RT(3)
where Rad is the radiance obtained by the target and RT is the radiance obtained by the white reference target.

The hyperspectral data organization, elaboration and processing described in this work was completed using MatlabR2021a (MahtWorks, Portola Valley, CA, USA).

### 4.5. Stress Induction and Evapotranspiration

Plants were grown in 3.2-litre pots containing 1.8 kg sand–peat mixture from the transplant to the end of the experiment. Water volume in the two theses, well-irrigated and stressed, respectively, was maintained at the field capacity (FC) [62] before the first stress induction. Drought stress stages were applied through a 70% reduction in irrigation water in stressed plants, maintaining 100% of the irrigation water in the control thesis. The exact water volume provided to the plant was assessed by a balance and irrigation control system installed on the conveyor belt. The water weight loss from pots was used to compute ET, defined as the water transpired daily by plants summed to water evaporated daily from pots and computed as:ET = TW(x) − TW(y)(4)
where TW(x) is the target weight on a day, and TW(y) is the target weight on the same day. The weight difference, expressed in grams (g), is the water evapotranspired by the plant.

### 4.6. Automatic Segmentation for the Hyperspectral Images

The application of an automatic segmentation method is needed to obtain a rapid plant pixel separation from the background. To achieve this objective, the background was covered with a matt black cover to delete reflection effects due to photons scattering. Segmentation’s first step was excluding the white target reference from the image. This operation was automated by constantly maintaining the reference position and target distance from the hyperspectral device during the acquisition process. According to this, the HI dimension was reduced to exclude the white reference panel. Segmentation’s second step was the application of a binary mask to separate the target from the background. The 69 images (one image per band with 2.5 nm spectral resolution) of the hyper-spectral cube were separately observed to visually select the specific spectral band where the difference between target and background was most evident. The reflectance value capable of separating the target from the background was chosen through a selection method based on defined reflectance values, selected through a percentile value selection. The suitable segmentation percentile value was determined based on the complete image background elimination and removal of geometric light reflection effects. Indeed, only plant regions in a perpendicular position to the hyperspectral camera optics should be selected. Binary segmentation masks were subsequently applied to images. The segmentation mask derived by one-band percentile segmentation was applied for all spectral bands.

### 4.7. Hyperspectral Data Analysis

After the calibration and segmentation of the HI, the HC was processed to extract the average signature by each pixel from each genotype at all acquisition times. The spectral signatures were processed to obtain the H-index capable of discriminating the stress status in genotypes during the experimental phases. The H-index was obtained by computing the maximum value of the approximate derivative point by point for each spectral signature. The result represents the maximum slope point in the acquired spectral range. The approximate derivative was performed using the diff function by MatlabR2021 (MahtWorks, USA). To differentiate control and stressed theses using the novel H-index, a total of 20 comparisons between theses and genotypes (one per genotype per five acquisition times) were performed using the ANOVA at both stress phases in addition to the recovery phase. The ANOVA was performed using the anova1 function by MatlabR2021a (MahtWorks, USA). The obtained results were correlated through a linear regression analysis with OIs to assess the relation between OIs and the H-index.

### 4.8. RGB Images Segmentation and OIs

The segmentation process and OIs retrieval were performed using the Python 3.8 with the PlantCV package v3.9 (https://plantcv.danforthcenter.org/, accessed on 12 November 2022). The first step consisted of image segmentation. Subsequently, pixel values were computed to obtain the following OIs:

#### 4.8.1. Projected Shot Area (PSA)

PSA is the sum of the number of pixels inside the plant region in each of the three digital RGB images taken at the three orthogonal views, and was computed in the following way [43]: PSA = TA(x) + TA(y)+ TA(z)(5)
where TA(x) is the target top area, TW(y) is the target side area and TW(z) is the target side area rotated 90 degrees; the result is the sum of the three areas expressed in cm^2^.

#### 4.8.2. HUE Index (HUE)

HUE is a single number corresponding to an angular position, from 0° to 360°, around a central point or axis on a colour wheel [44], where an angle of 0° corresponds to red, an angle of 90° corresponds to green, and an angle of 180° corresponds to the blue. Usually, leaf colour is included in a HUE range from 120° (green) to 60° (yellow). HUE is obtained by the mean of HUE pixel values inside the plant region, and was computed as follows:(6)$$\mathrm{HUE}=\frac{\sum_{1}^{n}PHV}{n}$$
where PHV is a single HUE pixel value and n is the pixel number inside the plant region.

#### 4.8.3. Senescence Index (SI)

SI is an optical index expressed as a percentage used to define a senescence status in plants, and it was computed as follows [45]:SI = (GAS − GerAS)/GAS(7)
where GAS is a green area in the side view, and the corresponding value is the sum of pixel values in the HUE angular region from 60° to 180°. GerAS is a greener area in the side view obtained by the sum of pixel values in the HUE angular region from 80° to 180°. 

### 4.9. Statistical Analysis

Optical (PSA, HUE and SI) and hyperspectral indices (H-index), measured at the end of each subcycle (stress and recovery periods) imposed to test the response to drought stress, were statistically analyzed separately for each genotype by means of one-way ANOVA (MatlabR2021a, MahtWorks, USA). 

## 5. Conclusions

In this study, four tomato genotypes were successfully tested in an HTPP LemnaTec Scanalzer 3D system during a tomato drought stress experiment. The HTPP was integrated for the first time with the low-cost Senop HSC-2 hyperspectral camera, and over 120 gigabytes of HIS were acquired and processed. The genotypes were tested during two cycles of full and partial irrigation, and the principal phenotyping traits, such as PSA, HUE and SI, were assessed. The RH and T parameters were collected, and VPD was obtained over the experiment. Therefore, based on the results obtained, the conclusions of this study are:(i)The integration of the low-cost hyperspectral camera in the HTPP based on the LemnaTec Scanalzer 3D system permitted an increase in the water stress detection capability by the HTPP; indeed, a hyperspectral camera with an NIR spectral range appears more efficient in detecting the stress status during the first stress and recovery phases than the RGB technologies.(ii)The PSA more efficiently describes the structural effect of the differential irrigation regimes, particularly during the first stress phase, than HUE and SI, where SI and HUE do not show a clear picture of the irrigation trend compared to the ET results. The H-index more clearly describes the ecophysiological response during the first stress and recovery phases compared to PSA, HUE and SI, showing a higher sensitivity and better reflecting the ET trend. In addition, the H-index more clearly represents the second stress phase than PSA, HUE and SI, which globally appear less visible, probably due to an adaptation to the stress condition by the stressed theses. Based on these results, a low-cost hyperspectral camera with a VIS/NIR spectral range integrated with the RGB technologies commonly used in phenotyping activities appears to be the optimal combination for adoption in the phenotyping process. In addition, this integration could open new ways to develop innovative phenotyping techniques.(iii)The segmentation method based on the percentile technique in a standardized acquisition set efficiently reduced the hyperspectral dataset dimension, reporting a data reduction of 85.5%; simultaneously, the time needed to process the HI was reduced.(iv)The tolerance and susceptibility to drought stress of the four genotypes were successfully assessed by the combination of optical and hyperspectral indexes. Overall, the OIs and H-index results show that genotypes 770P and 990P were more susceptible to water stress than Red Setter and Torremaggiore genotypes.

Finally, it is worth pointing out that the detection of different abiotic stress statuses, such as nutrient deficit and metabolic alteration, and pathologies caused by biotic stresses, need to be further explored. The results obtained in this study demonstrate new ways to use a low-cost hyperspectral camera in a HTPP, and further studies under different stress conditions using HI and exploring the SWIR spectral range are recommended.

## Figures and Tables

**Figure 1 plants-12-01730-f001:**
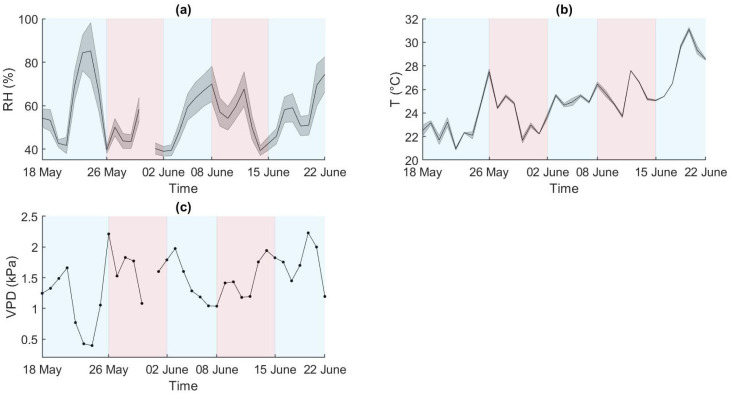
RH expressed in percentage (**a**), T expressed in °C (**b**), and VPD expressed in Kilopascal (**c**) trends during the experiment. The red and blue areas represent the stress and recovery periods respectively. (**a**,**b**) report the mean of two sensors and the associated standard deviation. RH data were not recorded on day 31 May.

**Figure 2 plants-12-01730-f002:**
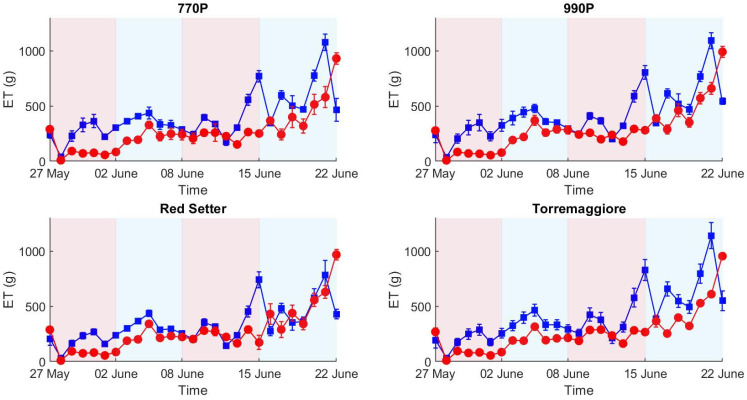
ET trend by 770P, 990P, Red Setter and Torremaggiore genotypes from the beginning to the end of the experiment. The red line represents the stressed thesis, and the blue line represents the control thesis. The red and blue areas in the plots represent the stress and full-irrigation experimental phases, respectively. Each point reports the mean of six samples and the associated standard deviation.

**Figure 3 plants-12-01730-f003:**
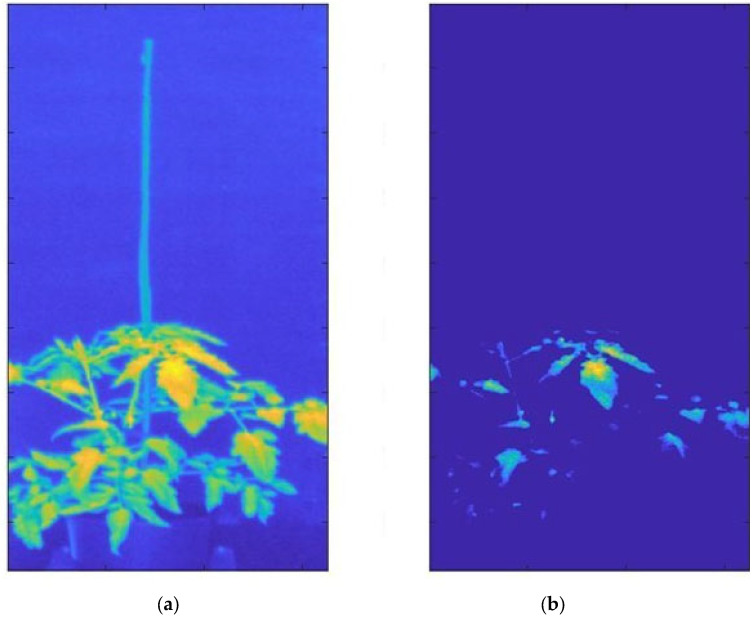
Example of segmentation results. The segmentation process based on the 96° percentile was applied to a Red Setter tomato genotype control thesis acquired on 8 June 2021. The original HI (**a**), composed of 1,048,576 pixels, was reduced to 11,408 pixels (**b**), permitting the sampling of the plant regions characterized by direct and orthogonal illumination.

**Figure 4 plants-12-01730-f004:**
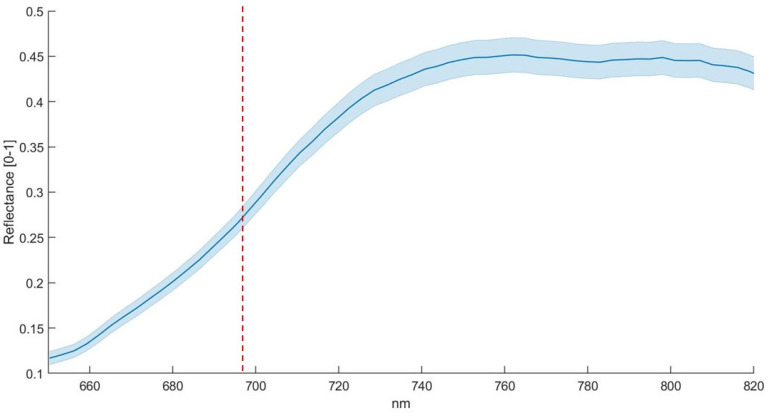
The spectral signature derived by the selected plant of Figure 3 after segmentation. The red dashed line shows the spectral band (691 nm) where the maximum value of the derivate was detected and used to derive the hyperspectral index (H-index). The blue line reports mean ± standard deviation of 11,408 pixels.

**Figure 5 plants-12-01730-f005:**
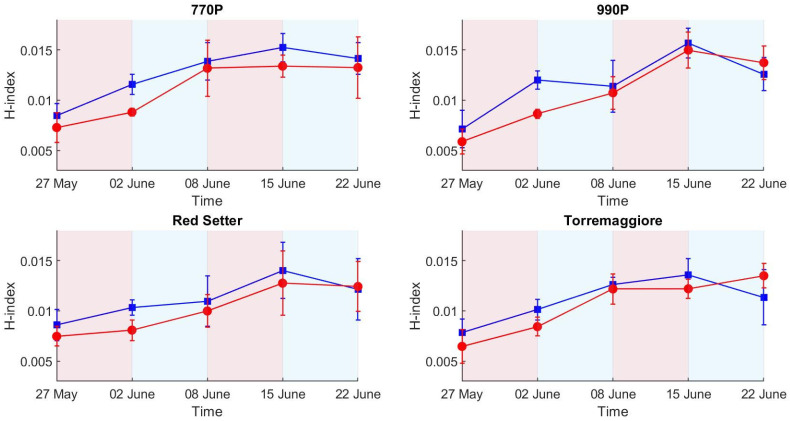
Hyperspectral index (H-index) values in stressed and control treatments (red and blue lines and points, respectively) for the four genotypes. Each point reports the mean of six samples and the associated standard deviation.

**Figure 6 plants-12-01730-f006:**
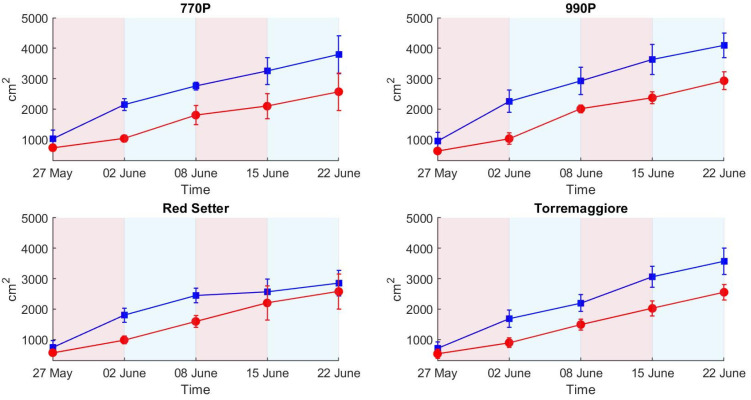
Projected shoot areas in stressed and control treatments (red and blue lines and points, respectively) for the four genotypes. Each point reports the mean of six samples and the associated standard deviation.

**Figure 7 plants-12-01730-f007:**
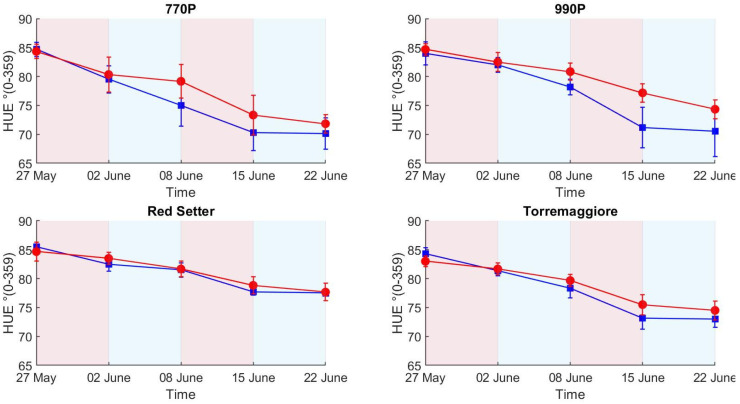
HUE index values in stressed and control treatments (red and blue lines and points, respectively) for the four genotypes. Each point reports the mean of six samples and the associated standard deviation.

**Figure 8 plants-12-01730-f008:**
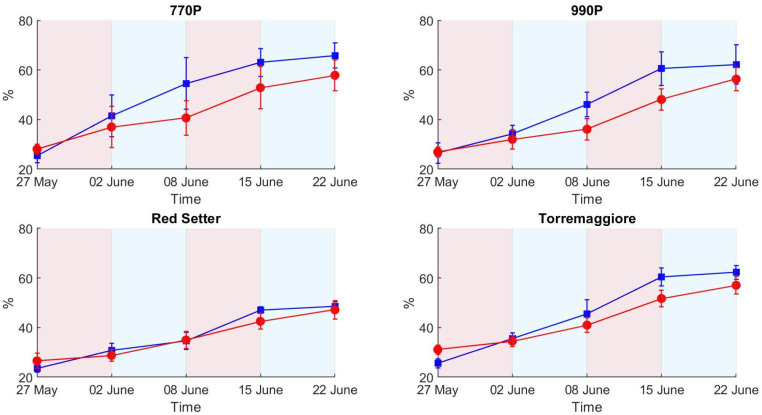
Senescence index values in stressed and control treatments (red and blue lines and points, respectively) for the four genotypes. Each point reports the mean of six samples and the associated standard deviation.

**Figure 9 plants-12-01730-f009:**
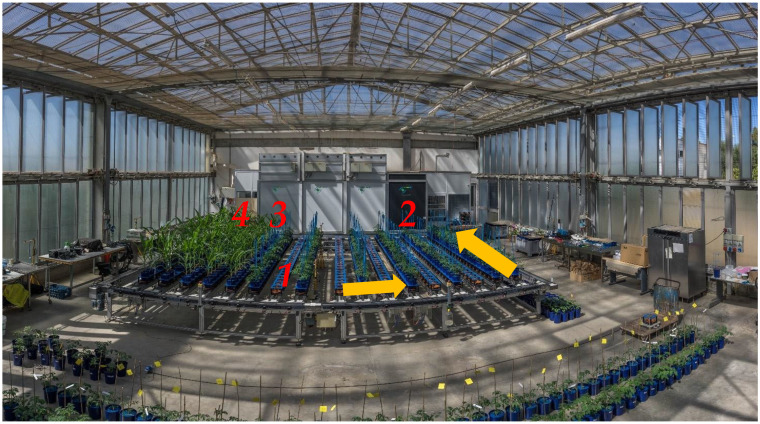
Plants located on the conveyor belt (**1**) were moved to visible camera (**2**) during the acquisition process. Subsequently, plants were moved to hyperspectral camera (**3**) and automatic irrigation system (**4**). Plants were left on the conveyor belt during the time between acquisition processes. The HTPP was based on a LemnaTec Scanalzer 3D system.

**Figure 10 plants-12-01730-f010:**
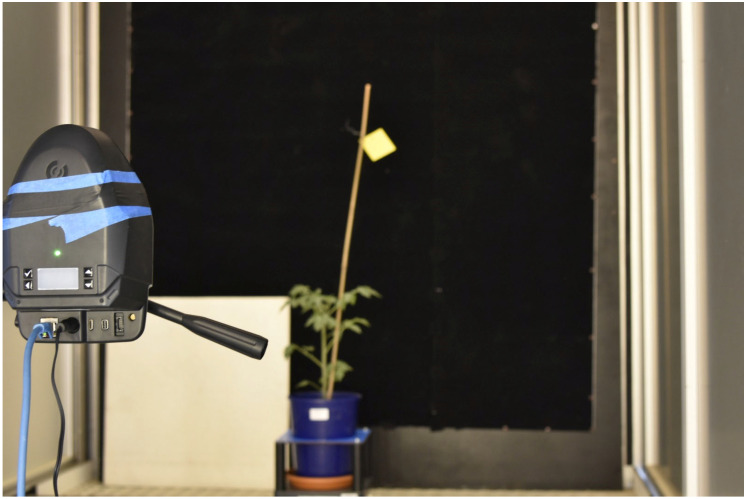
Hyperspectral acquisition process on tomato plant. The white Lambertian surface reference target at 75% of reflectance is the white panel on the plant’s left.

**Table 1 plants-12-01730-t001:** ANOVA results on hyperspectral index (H-index), projected shoot area (PSA), HUE index, senescence index (SI), calculated at the end of each sub-cycle period. The significance of the ANOVA was indicated using ** symbol.

		770P	990P	Red Setter	Torremaggiore
	H-index	**	**	**	**
I Stress (2 June)	PSA	**	**	**	**
	HUE	-	-	-	-
	SI	-	-	-	-
	H-index	-	-	-	-
I Recovery (8 June)	PSA	**	**	**	**
	HUE	-	**	-	-
	SI	-	**	-	-
	H-index	-	-	-	-
II Stress (15 June)	PSA	**	**	-	**
	HUE	-	**	-	-
	SI	-	**	**	**
	H-index	-	-	-	-
II Recovery (22 June)	PSA	**	**	-	**
	HUE	-	-	-	-
	SI	-	-	-	**

**Table 2 plants-12-01730-t002:** Genotypes tested during drought stress experiment.

Name	Provider	Origin	Plant Growth Habit	Fruit Type	Use
Red Setter	Portici Seed Collection	Commercial variety	Determinate	Elongated/Blocky	Processing
Torremaggiore	La Semiorto Sementi SRL	Southern Italy	Determinate	Round Cherry	Fresh Market
770 P	Portici Seed Collection	Southern Italy	Determinate	Round Cherry with apex	Processing
990 P	Portici Seed Collection	Southern Italy	Semi-Determinate	Round Cherry	Processing

**Table 3 plants-12-01730-t003:** Drought stress experiment timetable.

Operation	Day
Transplant	7 May 2021
RGB and Hyperspectral Images acquisition	27 May 2021
Start 1st stress	27 May 2021
RGB and Hyperspectral Images acquisition	2 June 2021
Start Recovery	2 June 2021
RGB and Hyperspectral Images acquisition	8 June 2021
Start 2nd stress	8 June 2021
RGB and Hyperspectral Images acquisition	15 June 2021
Start Recovery	15 June 2021
RGB and Hyperspectral Images acquisition	22 June 2021

## Data Availability

Not applicable.

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
