# Peer review of "Low-Cost Hyperspectral Imaging to Detect Drought Stress in High-Throughput Phenotyping"

_plants, 2023, doi:10.3390/plants12081730_

Round 1
Reviewer 1 Report
The present MS “Low-cost hyperspectral imaging to detect drought stress in high-throughput phenotyping” is an interesting Idea and written well. However, it still needs substantial changes before final decision.
The abstract section is poorly written poorly; in this section authors just presented the methodology right from starting to the end of this section. Therefore, I suggest the authors add the values of the study findings, what they have noticed and what are their outcomes.
Results sections: Relative humidity (RH) and temperature (T) data were collected within the HTP greenhouse environment for the duration of the experiment. The vapour pressure deficit (VPD) defined as the difference between the amount of moisture in the air and the amount the air can hold at saturation, was computed using RH and T data (figure 1). The daily maximum RH (RH max) was registered on 25/05, and the daily minimum RH 144 (RHmin) was registered on 14/06 (Fig. 1a). “No need to give this sort of information here in this section, authors must focus on the core findings and must write core findings in results section”. Check it throughout the results section of MS.
Line 160: control theses in all genotypes,??? Correct it, please.
The discussion section is written well, however, it needs more logical and cool reasoning supported by study findings.
Reviewer 2 Report
This manuscript (plants-2331860) integrates a low-cost hyperspectral camera into an HTP platform to evaluate drought stress resistance in tomato genotypes, introducing a highly capable hyperspectral index (H-index) to detect early-stage drought stress.
The topics as it presents an evaluation using low-cost equipment. I was only concerned about the 2.5 nm resolution; however, the measurements appear to be accurate.
In the abstract, I have included the main result and a concluding statement, rather than discussing "Finally, further applications of a low-cost hyperspectral camera in an HTP context were discussed." This allows readers to gain insight into the conclusions of your work.
Regarding the introduction, it could be shortened, as in some cases, I felt a bit lost with the amount of context provided. This might improve the initial flow of the manuscript.
As for the results and discussion, they are good.
Materials and methods are well described. Please kindly check the author guidelines for Plants, as corrections in citations and references to figures and captions are needed.
Suggestion:
Include the sample size in the figure captions and Mean±SE.
L73. high;
L92. Zea mays (italic);
L116., status
Table 2, need line in bottom;
Review the references list and replace sources dated prior to 2000 with more recent and relevant publications, where appropriate.
Round 2
Reviewer 1 Report
The authors have addressed my all comments therefore, paper can be accepted for publication.
Reviewer 2 Report
Dear Author, thank you for addressing my comments and suggestions. I recommend accepting this manuscript in its current form.